# Numerical Simulation Prediction of Erosion Characteristics in a Double-Suction Centrifugal Pump

**Xijie Song [1], Dunzhe Qi [2], Lijuan Xu [2], Yubin Shen [2], Wei Wang [1], Zhengwei Wang [1,*] and Yan Liu [1]**

[1] State Key Laboratory of Hydro Science and Engineering & Department of Thermal Engineering, Tsinghua University, Beijing 100084, China; songxijie@mail.tsinghua.edu.cn (X.S.); wwei2020@mail.tsinghua.edu.cn (W.W.); ly82@tsinghua.org.cn (Y.L.)

[2] Water Conservancy Project Construction Center of Ningxia Hui Autonomous Region, Yinchuan 750000, China; 18252739209@139.com (D.Q.); 15253410359@139.com (L.X.); 18795009750@139.com (Y.S.)

* Correspondence: wzw@mail.tsinghua.edu.cn

**Abstract:** The double-suction centrifugal pumps installed along the Yellow River in China face serious sediment erosion due to the high sediment content which causes the poor operation efficiency of the pump units. The particle motion characteristics and erosion characteristics in a pump under different flow rates and different particle concentrations were numerically simulated based on the particle track model of solid-liquid two-phase flow. The results show that the flow rate has a significant effect on the particle tracks and the erosion caused by the particles in the impeller. The total erosion rate is positively correlated with the flow rate, and increases with the increase in flow rate. The vortex and secondary flow in the impeller have obvious influence on the particle trajectory, which increases the particle concentration at the trailing edge of the pressure surface and intensifies the impact erosion in this area. The particles carried by the vortex intensifies the local erosion. The particle concentration mainly affects the erosion rate, but has little effect on the erosion position. The influence of flow rate on pump erosion is greater than that of the particle properties. These results provide a reference for optimization of the design of anti-erosion blades of double-suction pumps and the regulation and operation of pumping stations.

**Keywords:** centrifugal pump; impeller; erosion; particle track; particle concentration; flow rate

## 1. Introduction

Double-suction centrifugal pumps can produce large flow and high head, and are widely used in irrigation pumping stations along the Yellow River in China [1]. The Yellow River is one of the rivers with the highest sediment concentration in the world, which makes the pumping units from the pumping stations along the Yellow River suffer from serious sediment erosion, seriously affecting the normal operation of pumps, reducing the working efficiency of pumps, and causing huge positive economic losses and energy waste [2].

Up to now, a large number of studies on the solid-liquid two-phase flow in centrifugal pumps have been carried out, and in-depth studies on the erosion and damage on the parts of the flow passage of centrifugal pumps have also been conducted [3–5]. The erosion of different parts in the pump is mainly due to the change in the characteristics of two-phase flow and the characteristics of surface structure material, and the appearances of the damaged surfaces are different [6]. According to the erosion pattern in the pump, the erosion is divided into two basic patterns: general erosion and local erosion. The general erosion is caused by the particles in water flow in the pump scouring the surface of the flow passages [7]. The surface of the material with common erosion presents the same ripple erosion trace, which can clearly reproduce the movement direction of particles. Local erosion is more serious than general erosion which happens in some areas of the pump [8]. Local erosion is usually caused by the deterioration of the flow state near the

surface of the parts of flow passage [9]. Xu studied the movement of solid particles in a centrifugal pump impeller by using high-speed photography technology, and found that the particle size, the impeller speed, and the blade angle all have a significant effect on the movement of the particles [10]. Liu [11] found that the characteristics of discrete particles and impeller speed have important influence on both the trajectory of solid particles and the process of wall collision by tracking the particle trajectory in solid-liquid flow field. Qian found that the blade erosion rate is related to velocity and impact angle through numerical simulation of different blade inlet edge shapes [12]. K.C. Wilson et al. found that the main erosion mechanisms that cause pump erosion can be divided into sliding erosion and impact erosion, depending on the impact angle of the particles along the pump geometry [13]. Xu studied the flow in solid-liquid pump and found that the flow pattern of solid-liquid two-phase flow in solid-liquid pump is closely related to erosion. The better the flow condition in the pump (no large-scale flow separation, local high velocity and large pressure pulsation, the smoother the flow is), the less the degree of erosion in the flow passages [14]. At different stages of the operation life, the erosion rate of solid-liquid pump varies greatly. In the early stage of erosion, the erosion on the flow passage caused by particles is similar to polishing on the surface, and the speed of erosion is slow and the degree of erosion is small. In the stable erosion period, the erosion of the flow passage parts is mainly general erosion, and the degree of the local erosion increases gradually [15]. The erosion causes the deterioration of the flow state in the pump, and then accelerates the local erosion. In the later stage of erosion, the erosion is further accelerated, the material loss of the parts in the flow passages increase more than that of the previous two stages, and cavitation may also be induced. The combination of erosion and cavitation greatly aggravates the damage of the materials until the parts in the flow passage are damaged. When the same slurry is transported by different pumps and different slurry is transported by the same pump, if the operation efficiency of the pump is high, the erosion is uniform [16]. If the operation efficiency of the pump is low, the serious erosion caused by impact and vortex will occur. After the blade outlet is worn, the blade will become thinner. Especially in the root of the hub, the erosion is serious, and the strength of the blade is weakened. Finally, the blade will break because it cannot bear the pressure difference between the pressure surface and the suction surface [17].

The particles are taken as discrete phase and liquid as continuous phase in the particle track model, and particle motion in Lagrange coordinate system and continuous phase motion in Eulerian coordinate system are calculated. Then, a large number of particles are counted so that the macro trajectory of particle motion is obtained [18]. The velocity of particles on any trajectory can be obtained by using Lagrange method. Xu used this method to simulate the movement of solid particles in the centrifugal pump, and the predicted results are close to the experimental values [19]. Meanwhile, in order to ensure the reliability of erosion rate prediction, Finnie [20] and Tabakoff [21] proposed different erosion rate prediction models. These empirical models are basically established on the basis of erosion tests, and the actual variables (impact velocity, impact angle and impact frequency) determine the size of these variables in these empirical models. Some studies prove that Tabakoff erosion model is more accurate [22,23].

In this paper, the erosion of a double-suction centrifugal pump along the Yellow River is predicted and analyzed by numerical simulation due to the large seasonal variation of the Yellow River water flow. This paper focuses on the study of particle trajectory and erosion distribution in the impeller of double-suction centrifugal pump under different flow rate conditions and different sediment concentrations. Tabakoff erosion model is used to predict the erosion of centrifugal pump. This study is helpful to reveal the mechanism of erosion and solve the problem of erosion damage in engineering.

## 2. Materials and Methods

### 2.1. Research Object

The research object of this paper is a double-suction centrifugal pump. The main parameters of the entire calculation model are as follows: rated flow $Q_d$ = 3.083 m$^3$/s, rated head H = 50 m, rated efficiency $\eta_r$ = 86%, rotation speed $n$ = 490 r/min, number of blades $z_b$ = 16, impeller diameter $D$ = 1275 mm, as shown in Figure 1. Figure 2 shows the name of the impeller components, $\omega$ is rotational angular velocity.

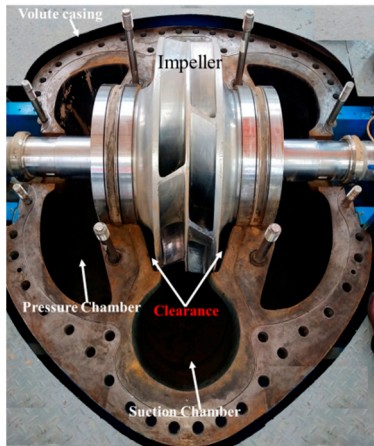

**Figure 1.** Centrifugal pump.

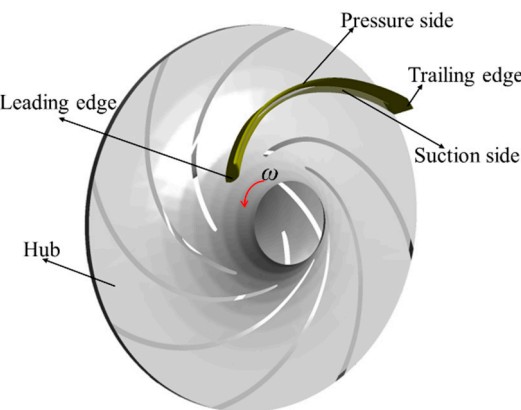

**Figure 2.** Impeller components.

The calculation model is meshed by the software Workbench, and the unstructured grid and structured grid are used, as shown in Figure 3, and the grid independence check is carried out by analyzing the change in head under different the number of grids, as shown in Figure 4. When the number of grids reaches 4.21 million, the head basically remains unchanged, so the final number of grids is 4.21 million.

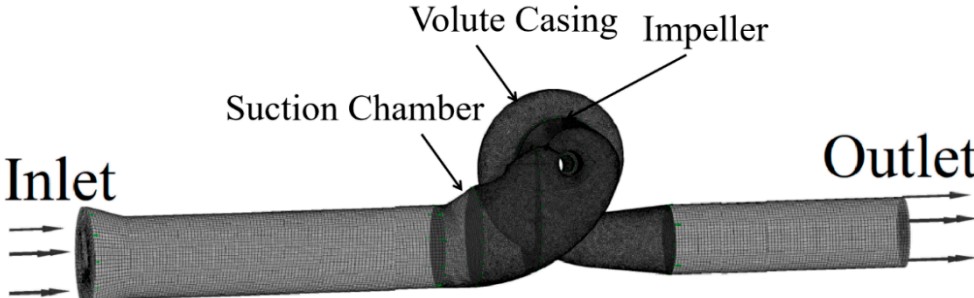

**Figure 3.** Grid of the calculation model.

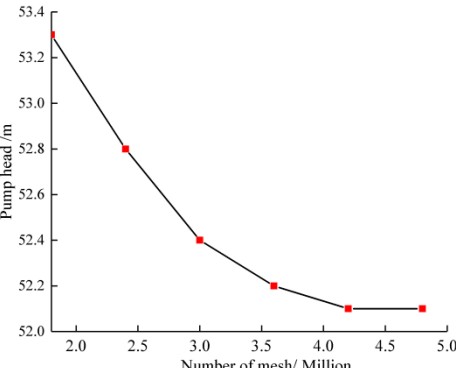

**Figure 4.** Change curve of the pump head with grid number.

### 2.2. Basic Equation of Particle Motion

In the Lagrangian framework, when the particles move in the liquid, the force comes from the velocity difference between the particles and the fluid.

The main forces on particles are gravity, resistance, virtual mass force, pressure gradient force, Basset force, Saffman force, Magnus force, and so on [24].

$$m_p \frac{du_p}{dt} = F_D + F_B + F_G + F_V + F_P + F_X \tag{1}$$

where $t$ is time, $m_p$ is particle mass, $u_p$ is particle velocity, $F_D$ is resistance, $F_B$ is basset force, $F_G$ is gravity, $F_V$ is virtual mass force, $F_P$ is pressure gradient force, $F_X$ is the sum of other external forces considered.

In this paper, the particle concentration in the flow field is small, the fluid velocity of the continuous phase in the pump is large, and there is a large density difference between the continuous phase and the discrete phase. Therefore, the virtual mass force, pressure gradient force, basset force, Saffman force, and Magnus force on the solid particles can be ignored. The basic equation of particle motion can be expressed as:

$$\frac{dx_{pi}}{dt} = u_{pi} \tag{2}$$

$$\frac{du_{pi}}{dt} = \frac{3C_D \rho f}{4\rho_p D_p} |u_s| u_s \tag{3}$$

where $u_s$ is the slip velocity between particles and liquid, $C_D$ is the drag coefficient related to Reynolds number, $\rho_f$ is the liquid density, $\rho_p$ is the particle density, $D_p$ is the particle diameter, and $x_{pi}$ is the spatial coordinate position of particles. It can be seen from the formula that when the particle moves in the liquid, the particle trajectory is related to the particle diameter and density.

### 2.3. Erosion Model

ANSYS CFX is used to calculate the flow field. The particle track model is used to calculate the flow field. The coupling mode of the calculation model is one-way coupling. Tabakoff erosion model is used to predict erosion. According to the particle track model, each group of particles moves along its own independent trajectory from the initial position, the particles are independent of each other, and there is relative velocity slip between particles and fluid, ignoring the turbulent diffusion, viscosity, and heat conduction between particles. Tabakoff erosion model is an empirical and semi-empirical erosion model considering the effects of different particle velocities and collision angles on target erosion.

The model is based on the impact angle and collision velocity (i.e., particle trajectory) of particles impacting the surface.

$$E = f(\gamma)\left(\frac{V_p}{V_1}\right)^2 \cos^2\gamma\left[1 - \left(1 - \frac{V_p}{V_3}\sin\gamma\right)^2\right] + \left(\frac{V_p}{V_2}\sin\gamma\right)^4 \tag{4}$$

$$f(\gamma) = \left[1 + k_1 k_{12}\sin\left(\gamma\frac{\pi/2}{\gamma_0}\right)\right]^2 \tag{5}$$

$$k_1 = \begin{cases} 1 \ \gamma \le 2\gamma_0 \\ 0 \ \gamma > 2\gamma_0 \end{cases}$$

Here, $E$ is the dimensionless mass (mass of eroded wall material divided by the mass of particle). $V_p$ is the particle impact velocity. $V_1$, $V_2$, and $V_3$ are the parameters of particle impact velocity. $\gamma$ is the impact angle in radians between the approaching particle track and the wall, $\gamma_0$ being the angle of maximum erosion. $k_1$, $k_{12}$, and $\gamma_0$ are model constants and depend on the particle/wall material combination.

Formula (4) can be divided into two parts: the first is the small angle cutting damage of particles, which is the damage mechanism of particles on the ductile materials; the second is the erosion damage of the target by the normal velocity of particles, which is proportional to the Fourth Square of the velocity, which is the failure mechanism of particles on brittle materials.

Because the erosion model can comprehensively consider the joint effects of ductile materials and brittle materials, it can more comprehensively predict the erosion characteristics. However, due to the large number of empirical coefficients and strong pertinence, the erosion model is mainly suitable for steel, aluminum, and other materials.

### 2.4. Calculation Method for Solution

The total pressure inlet condition is adopted in the calculation domain, and the volume fraction of particles at the inlet is assumed to be evenly distributed. The total mass flow rate is adopted at the outlet. For the wall in contact with liquid in the parts in the flow passages, the non-slip wall condition is adopted for the fluid phase, the free slip wall condition is adopted for the solid particle phase, and the standard wall function is adopted near the wall. The SST k-ω was chosen as the turbulence model, the particle track model was used for solid phase, and Tabakoff erosion model was used for erosion model. High precision difference scheme and RMS residual scheme are used to solve the problem, and the accuracy is set to $10^{-5}$.

## 3. Discussion of Calculation Results

### 3.1. Verification of Calculation Scheme and Reliability

In this paper, the particle motion under different flow rate and concentration was analyzed. In the CFX setting, the discrete phase of particles was set as dilute phase, and the collision between particles was not considered. The collision between the particle and the solid wall was assumed to be completely elastic without considering the energy loss. The characteristics of flow and erosion inside the impeller were analyzed under three typical conditions of $0.25Q_d = 0.771 \text{ m}^3/\text{s}$, $Q_d = 3.083 \text{ m}^3/\text{s}$ and $1.27Q_d = 3.91 \text{ m}^3/\text{s}$, as shown in Table 1.

**Table 1.** Sediment condition.

| Parameter | Scheme Number | | | | | |
|---|---|---|---|---|---|---|
| | **1** | **2** | **3** | **4** | **5** | **6** |
| Flow rate (m³/s) | $0.25Q_d$ | $Q_d$ | $1.27Q_d$ | $Q_d$ | $Q_d$ | $Q_d$ |
| particle concentration (kg/m³) | 15 | 15 | 15 | 11 | 7 | 1 |

The energy performance curve of the double-suction centrifugal pump is obtained by clear water calculation, as shown in Figure 5. The experimental results are provided by Andritz company. The energy performance obtained by numerical simulation is consistent with the experimental results, and the error is within 0.3%.

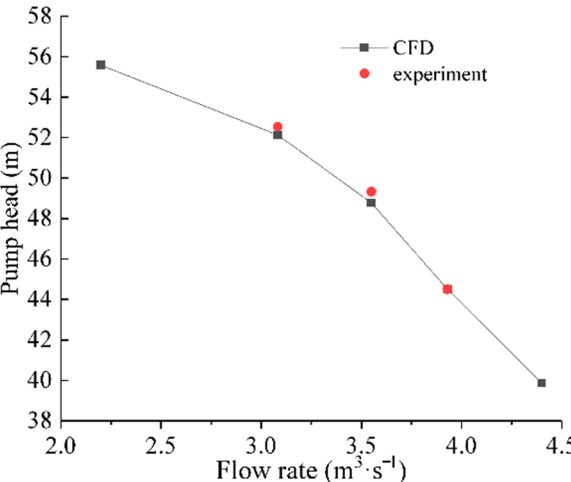

**Figure 5.** Change curve of pump head.

Figure 6 is a physical picture of the erosion of the pump unit of the Yellow River pumping station. Figure 7 shows the distribution of erosion rate when the particle inlet particle concentration is 15 kg/m$^3$ and the particle size is 0.025 mm. In engineering, pump erosion is due to long-term damage caused by different factors. However, numerical simulation is an ideal method to calculate the erosion prediction under a specific condition, and there are inevitably some errors in the predicted erosion results. On the whole, the erosion position and erosion morphology of the front cover plate wall and the blade outlet pressure surface predicted by the numerical simulation were consistent with the erosion characteristics of the pump impeller in the project site. The calculation results of external characteristics and erosion show that the numerical simulation method is reliable.

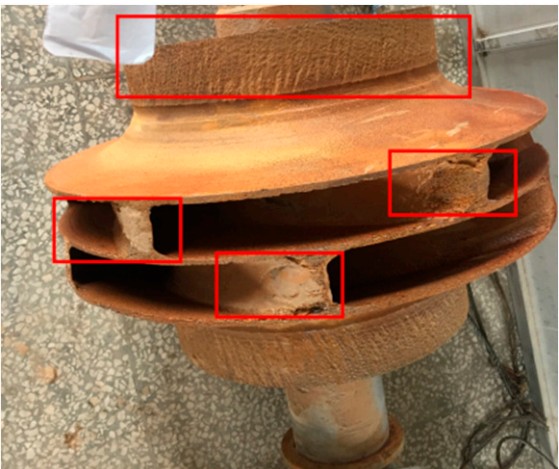

**Figure 6.** Physical picture of erosion.

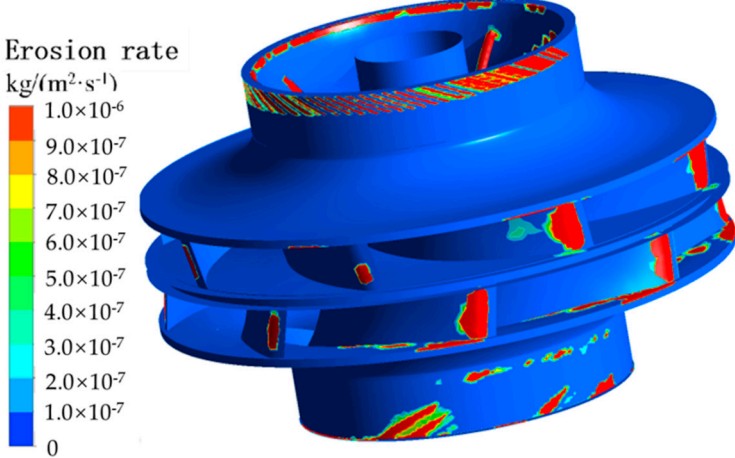

**Figure 7.** Distribution of erosion rate (CFD).

*3.2. Analysis of Flow and Erosion in Double-Suction Pumps under Different Flow Conditions*

3.2.1. Influence of Flow Rate on Particle Movement

In the process of pump operation, it often needs to operate under different flow conditions. The change in flow conditions often leads to the change in flow field structure and particle distribution in the pump. This section focuses on the influence of flow rate on particle trajectory and particle distribution. Figure 8 shows the flow pattern distribution on the surface of impeller cover plate under different conditions when the particle concentration is 15 kg/m$^3$. Under the condition of small flow rate, the flow pattern in the impeller was the worst, and there were different degrees of vortex and secondary flow caused by flow separation in many locations, as shown in Figure 8a. The multi-scale vortex flow in the blade channel blocks the blade channel, and the junction between the vortex boundary and the blade was the high-speed region. There were frequent frictions between the particles carried by the vortex and the wall. With the increase in flow rate, the uniformity of flow pattern in the impeller was obviously improved. The flow velocity in the pump under the rated flow condition $Q_d$ is relatively small compared with that under the large flow rate condition 1.27$Q_d$ and due to the existence of the outlet tongue, so a small-scale vortex appears near the tongue, as shown in Figure 8b. Under the two flow conditions of 1.27$Q_d$, the flow was smooth in the blade channel, and there was no vortex, as shown in Figure 8c.

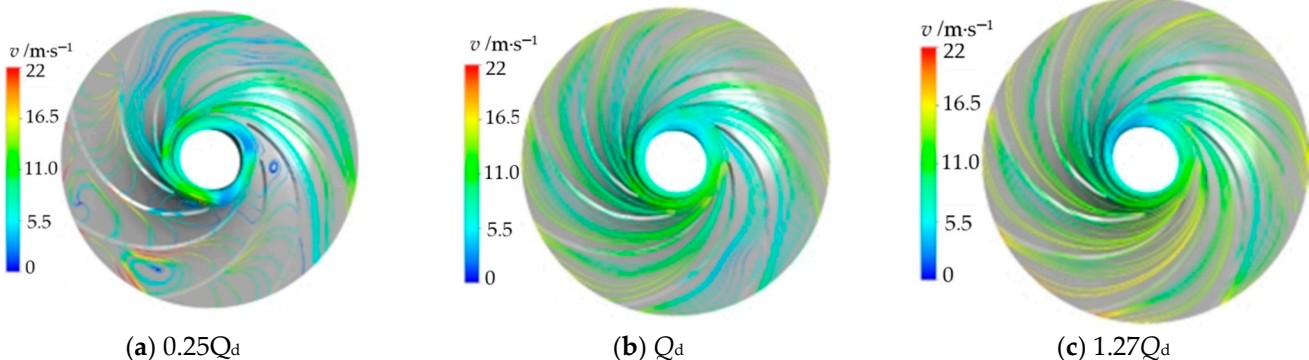

**(a)** 0.25$Q_d$　　　　　　　　　**(b)** $Q_d$　　　　　　　　　**(c)** 1.27$Q_d$

**Figure 8.** Flow pattern in the impeller under different flow rate conditions (15 kg/m$^3$).

Figure 9 shows the particle distribution in the impeller under different conditions when the particle concentration is 15 kg/m$^3$. Under the condition of small flow, the particle distribution in the impeller was uneven, as shown in Figure 9a. With the increase in flow, the uniformity of particle distribution in the impeller increased. However, the number of particles and the relative velocity of particles varied greatly in different blade channels. At

the impeller inlet, the particle concentration distribution was the largest, and the number of particles in contact with the impeller inlet was also the largest. The relative velocity of the particles at the tail edge of impeller outlet was the largest under the non-rated condition (1.27$Q_d$), as shown in Figure 9c. The distribution patterns of the particle in the impeller were consistent with the changes in the flow states under different flow conditions, which indicates that the flow rate has a great influence on the characteristics of the particle movements.

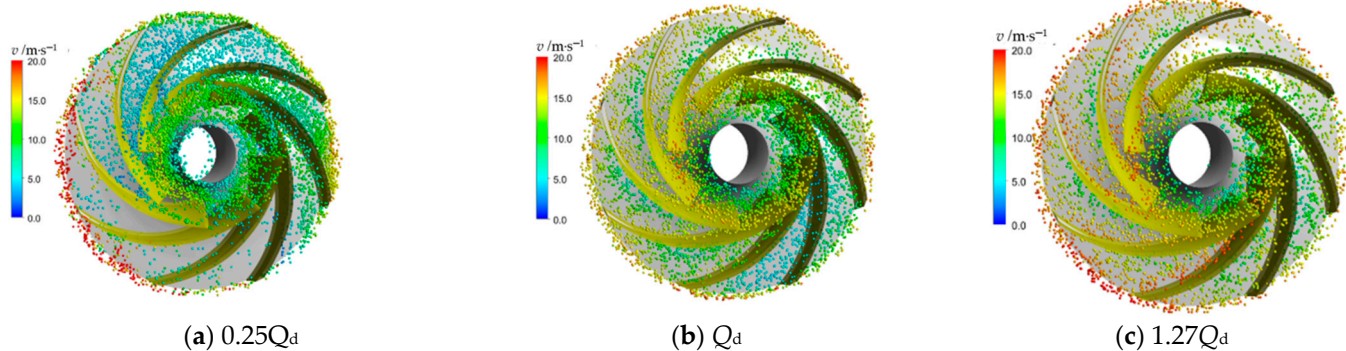

(**a**) 0.25$Q_d$       (**b**) $Q_d$       (**c**) 1.27$Q_d$

**Figure 9.** Particle distribution in the impeller under different flow rate conditions (15 kg/m$^3$).

### 3.2.2. Effect of Flow Rate on Surface Erosion

The erosion characteristics under different flow rate conditions are analyzed. In the definition of erosion rate, there are different forms of definitions, mainly including time erosion rate, area erosion rate, and volume erosion rate. For the wall erosion, the area erosion rate is mainly used to characterize the erosion characteristics, and the total erosion amount is the time erosion rate [25]. The calculation formula of area erosion rate is shown in Equation (6), and the calculation formula of time erosion rate is shown in Equation (7).

$$I_s = dW/d(S \times T) \tag{6}$$

where $I_s$ the area erosion rate, kg·m$^{-2}$·s$^{-1}$, W is the erosion wight, kg, $S$ is the erosion area, m$^2$, $T$ is the erosion time, s.

$$I_T = dW/dT \tag{7}$$

where $I_T$ is the erosion rate within a certain period of time, kg·s$^{-1}$, W is the erosion wight, kg, m$^2$, $T$ is the erosion time, s.

Figures 10 and 11 show the erosion distribution in the impeller under different flow rate conditions when the particle concentration is 15 kg/m$^3$. The results show that the erosion distribution and erosion mass loss of blades and hub on both sides of double-suction centrifugal pump are asymmetric. The erosion positions under different flow conditions were obviously different, which was directly related to the distribution position and motion trajectory of particles, indicating that the flow condition directly affected the erosion positions in the impeller. Under the condition of small flow rate, the internal erosion of double-suction centrifugal pump mainly included inflow impact erosion, blade channel friction erosion, vortex erosion, and blade head impact erosion, as shown in Figures 10a and 11a.

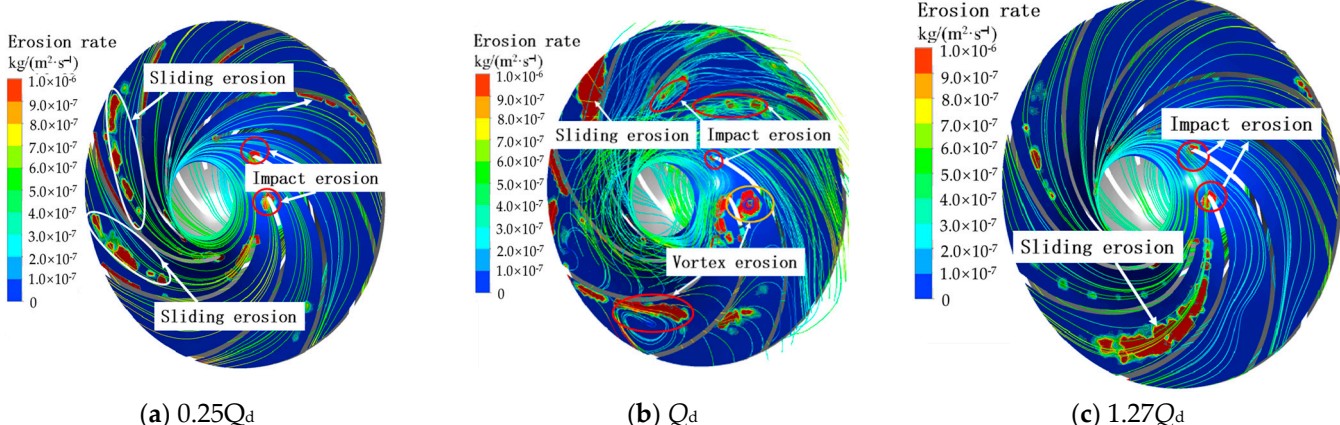

**Figure 10.** Erosion distribution of hub under different flow rate conditions (15 kg/m³).

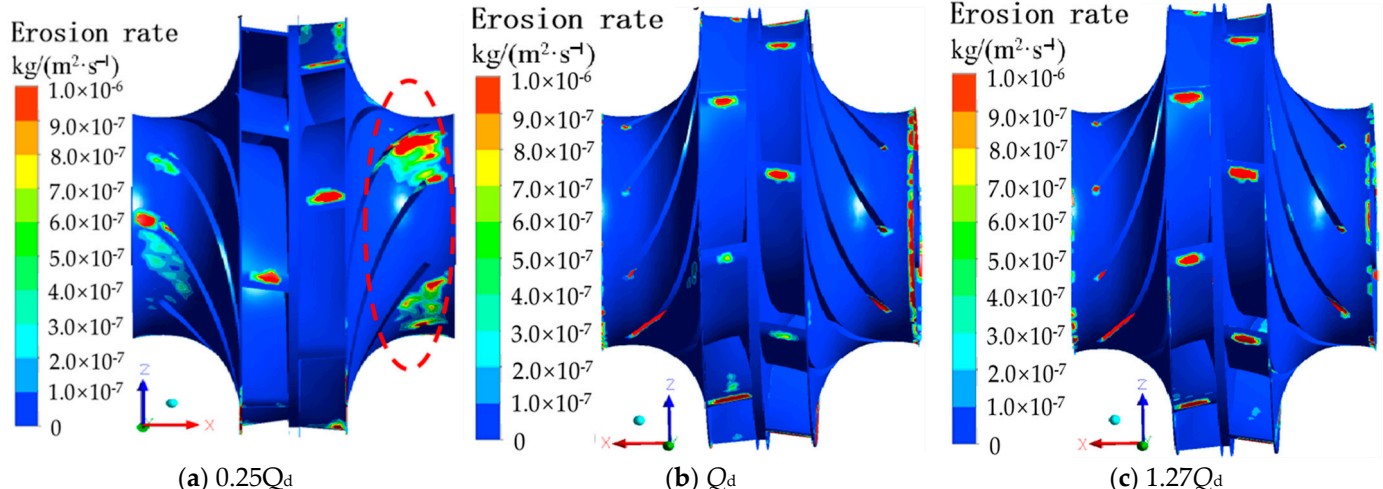

**Figure 11.** Distribution of erosion in the impeller under different flow rate conditions.

With the increase in flow rate, the inflow gradually became uniform, and the inflow impact erosion and vortex erosion disappeared, as shown in Figure 10b,c. The internal erosion of impeller was mainly the friction erosion in the blade channel and the impact erosion at the blade head. With the increase in flow rate, the erosion range in the blade channel increased gradually, and the characteristics of anti-erosion in the impeller were the best under the rated condition. The larger the flow rate, the more uniform the particle distribution, and the smaller the erosion degree on the hub and blade passage wall, but the greater the particle concentration at the impeller inlet, the higher the relative velocity, and the higher the erosion intensity on the blade head.

Through the analysis, it can be seen that the blade wall and the hub wall were worn under different flow conditions, which belonged to the common erosion caused by flow.

Under the condition of small flow rate, there was local erosion caused by vortex, and the local erosion caused by vortex was mainly friction erosion. The total erosion rate of the inner blade and hub wall under different flow conditions was analyzed quantitatively. Figures 12 and 13 show the total erosion rate curves of blade wall and hub wall under different flow conditions. The total erosion rate of the blade wall positively correlated with the flow rate, and increased with the increase in flow rate. The results show that the erosion rate in the impeller is obviously different under different flow conditions, which indicates that the flow condition not only affects the erosion position in the impeller, but also has a great influence on the erosion rate. The results show that there are differences in the variation curves of the wall erosion rate of the blades on both sides, which indicates that the total erosion amount of the blades on both sides is different.

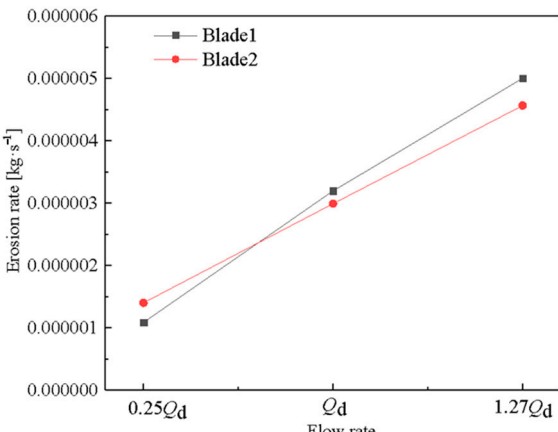

**Figure 12.** Erosion rate at blade changes with flow rate.

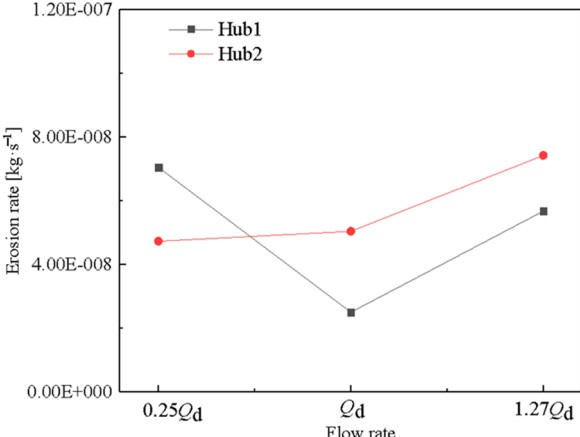

**Figure 13.** Erosion rate at hub changes with flow rate.

The erosion rate of the hub wall decreases from $0.25Q_d$ to $Q_d$ and increases from $Q_d$ to $1.27Q_d$. The variation of the erosion rate of the hub wall with the flow condition also shows that the erosion amount on both sides of the double-suction centrifugal pump is also different.

### 3.2.3. Local Erosion Caused by Vortex in Impeller

In this section, the erosion mechanism under the condition of small flow is analyzed, focusing on the vortex structure and the local erosion caused by second flow. Figure 14 shows the three-dimensional structure of the vortex in the double-suction centrifugal pump using $Q$ criterion. Due to the uneven velocity distribution in the impeller blade passage, the vortex attached to the hub and the passage vortex along the blade passage are generated in the impeller. The vortexes have a great influence on the particle trajectory. Under the influence of the flow passage vortex, the particle trajectory moves from the suction to the pressure surface in the outlet section of the impeller. Under the influence of passage vortex, the particle concentration on the pressure surface at the outlet of impeller increases, which aggravates the erosion in this area. The particles carried by the local wall-attached vortex rub against the wall of the local vortex region, which aggravates the local friction and erosion. Under the condition of small flow rate, the flow field structure in the impeller is unstable and the main flow velocity is relatively low, which leads to the secondary flow in the impeller. Due to the secondary flow in the blade channel, the particles are easier to separate from the carrier, causing the local velocity increase. With the increase in local velocity, the relative velocity of particles will also increase.

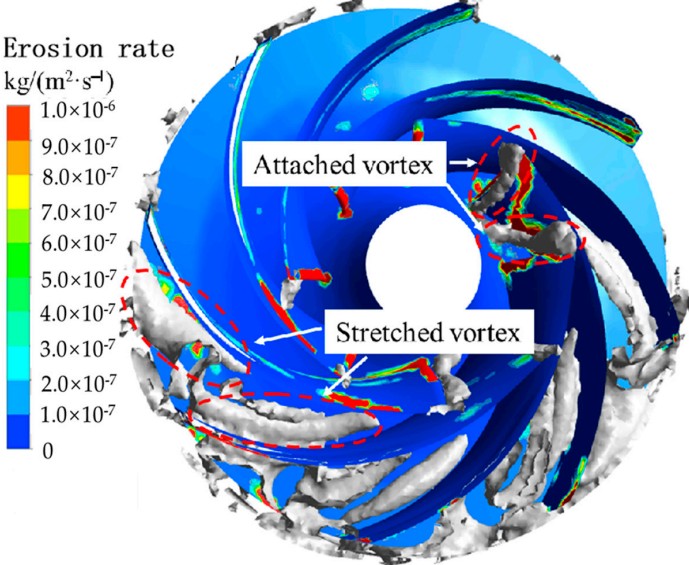

**Figure 14.** Vortex structure in double-suction centrifugal pump.

In addition, compared with the back cover shell, the velocity and vortex intensity near the blade surface are higher, and the particles are easier to impact the blade surface, so the asymmetric erosion on the blade surface is more obvious. The vortex erosion under small flow conditions is a local erosion, and the vortex rotation speed reflects the vortex rotation intensity and reflects the degree of the local erosion caused by vortex.

According to the previous analysis, the erosion degree of the vortex on the wall along the radial direction of the vortex is different, so the relationship between the erosion rate and the radius of the vortex core is used to quantitatively analyze the correlation between the erosion rate and the vortex. The variation curves of the tangential velocity of vortex core and the surface erosion rate at vortex area with the radius of vortex core are shown in Figure 15. The results show that the tangential velocity of vortex core increases with the increase in the vortex core radius, and the maximum tangential velocity occurs at the maximum vortex core radius. The results show that the erosion rate at the vortex core boundary is the largest; this is because the particle concentration near the vortex core center is low, the particle relative velocity is small, and the erosion rate is small. However, the vortex moves in the impeller passage continuously, and the vortex boundary will be subject to uneven friction of particles carried by the vortex, resulting in local erosion. The greater the radius of vortex core, the greater the rotational speed of vortex and the higher the erosion rate.

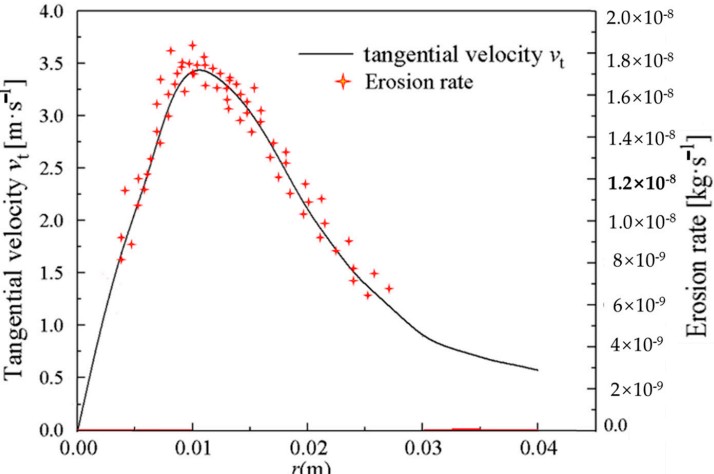

**Figure 15.** Tangential velocity and erosion rate change with vortex core radius.

### 3.3. Effect of Particle Concentration on Erosion Characteristics

Generally, the water pump operates in rated condition for a long time. The particle movement trajectory and erosion distribution under different particle concentrations under the rated flow conditions $Q_d$ are analyzed to explore the influence of particle concentration on erosion characteristics.

#### 3.3.1. Influence of Particle Concentration on Particle Tracks

Figure 16 shows the trajectory of a single particle with different particle concentrations. It can be seen that the change in particle concentration has little effect on the trajectory of particles. This is because the particle size does not change, and its force state does not change. Therefore, when the particle concentration changes, its trajectory will not change significantly.

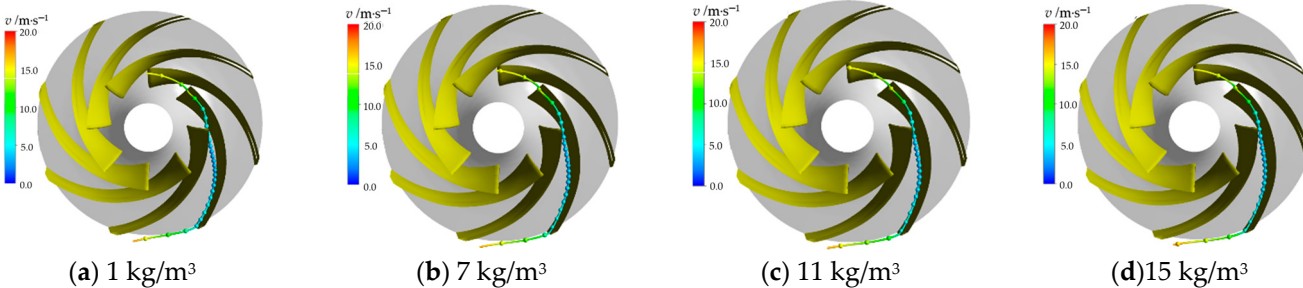

(**a**) 1 kg/m³   (**b**) 7 kg/m³   (**c**) 11 kg/m³   (**d**)15 kg/m³

**Figure 16.** Trajectory of single particle with different particle concentration.

Figures 17 and 18 show the particle distribution and solid volume fraction distribution under different particle concentrations, respectively. In the case of the same particle size, the distribution of particles in the centrifugal pump under different particle concentrations does not change obviously, so the particle concentration has little effect on the distribution of particles in the single channel, which is consistent with the description of particle trajectory. After the sediment laden flow enters the impeller, the particles accumulate at the leading edge of the blade. When the particle concentration is 1 kg/m³, the particle distribution in the impeller is very uniform, and there is a strip of particle aggregation distribution at the impeller inlet, as shown in Figure 17a. With the increase in particle concentration, particles gather in the outlet direction of the suction surface of the blade passage, and the impact frequency of particles on the blade surface increases. The larger the particle concentration is, the more uneven the particle distribution is in the impeller, and the more serious the particle aggregation is on the suction surface and pressure surface of the blade, as shown in Figure 17b–d. Due to the backflow near the tongue of the impeller, the concentration of particles near the tongue increases.

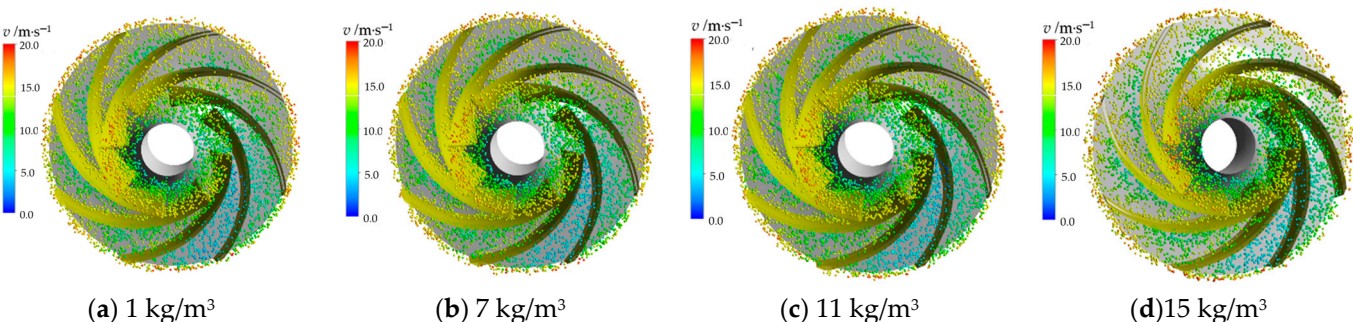

(**a**) 1 kg/m³   (**b**) 7 kg/m³   (**c**) 11 kg/m³   (**d**)15 kg/m³

**Figure 17.** Particle distribution under different particle concentrations.

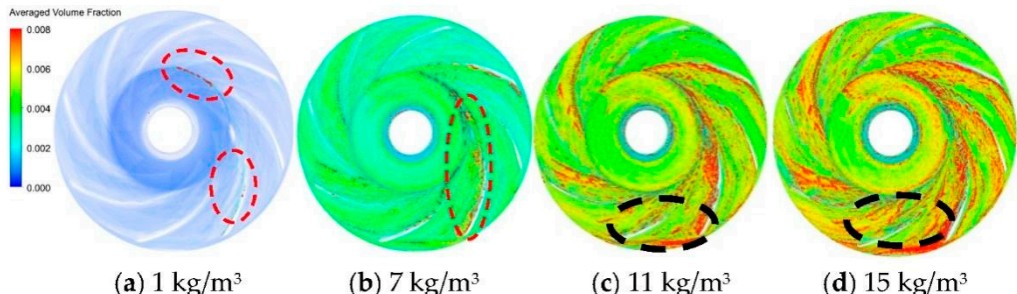

**Figure 18.** Distribution of solid volume fraction under different particle concentrations.

### 3.3.2. Effect of Particle Concentration on Erosion Characteristics

Figure 19 shows the distribution of single blade erosion rate with different particle concentrations. It can be seen from Figure 19 that the erosion rate of leading edge and trailing edge increases with the increase in particle concentration under the same particle size and different particle concentration, and the erosion morphology is the same. It can be seen that the change in particle concentration basically does not change the erosion morphology and erosion position— it only changes the erosion range and erosion rate on the original position, which is consistent with the particle motion trajectory, that is, the particle concentration has little effect on the erosion morphology and erosion position.

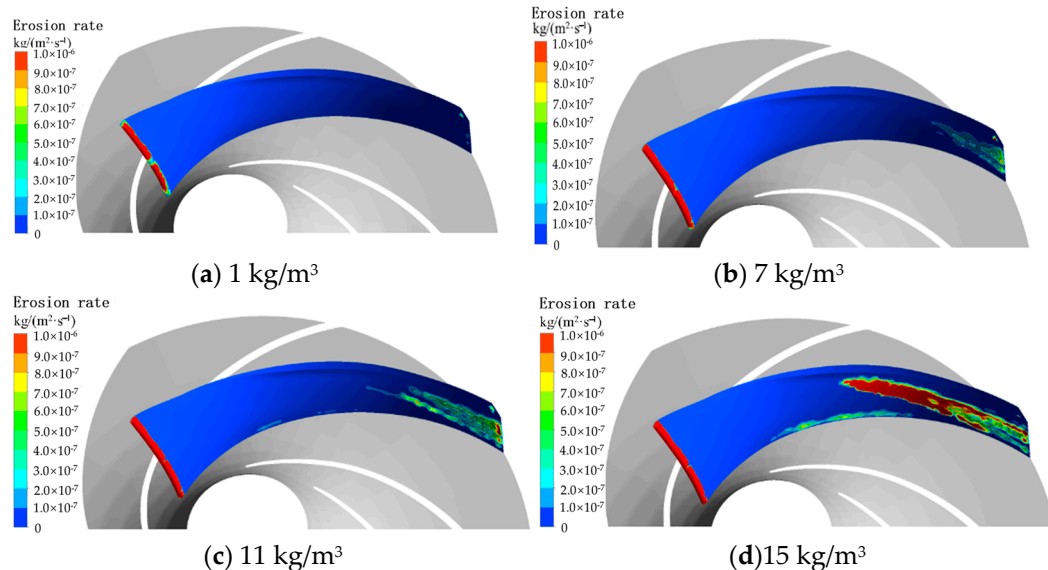

**Figure 19.** Erosion rate distribution of single blade with different particle concentrations.

To quantitatively analyze the erosion rate with the change in particle concentration, assuming that the pump runs continuously for one year, the changes in erosion mass loss at different positions of a single blade with particle concentration are shown in Figure 20. The results show that the erosion mass loss at different positions increases with the increase in particle concentration, which indicates that the particle concentration has a great influence on the erosion rate, which is consistent with the previous description. Under the same particle concentration, the erosion mass loss at the leading edge of blade is the largest, and that of blade tail is the smallest. The erosion mass loss at pressure surface is larger than that of suction surface, which is consistent with the particle motion characteristics and erosion distribution described in the previous paper [26].

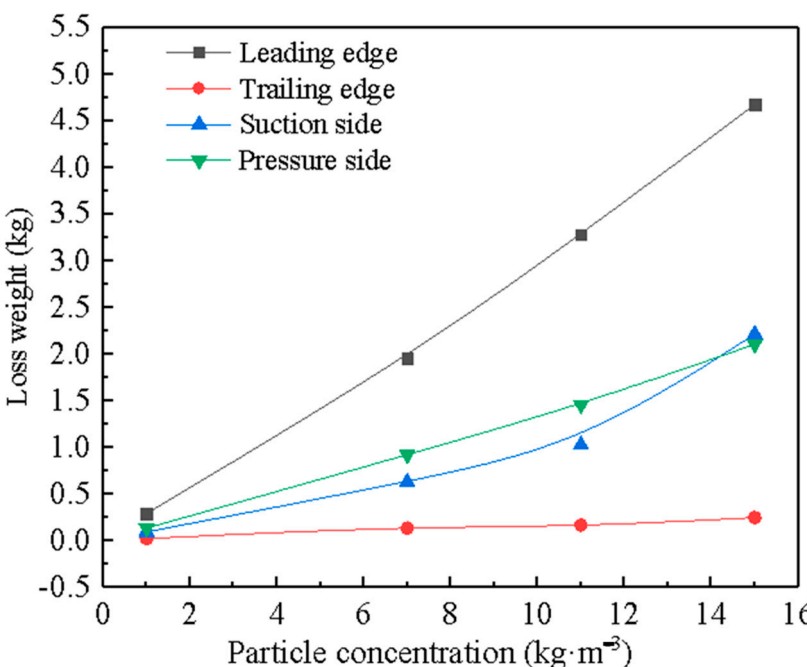

**Figure 20.** Change curve of erosion mass loss of single blade with different particle concentration.

## 4. Conclusions

Sediment erosion is a major problem affecting the performance and operation life of double-suction centrifugal pumps. This paper analyzes the particle distribution, particle trajectory, and surface erosion in the impeller under different flow conditions, $0.25Q_d$, $Q_d$, and $1.27Q_d$, and different particle concentrations, $1 \text{ kg/m}^3$, $7 \text{ kg/m}^3$, $11 \text{ kg/m}^3$, and $15 \text{ kg/m}^3$. The conclusions are as follows:

(1) The movements and erosion characteristics of particles in the impeller under different flow conditions are revealed. The flow structure in the double-suction centrifugal pump has a great influence on the erosion in the pump. The flow rate affects the particle trajectory and the uniformity of particle distribution. The more uneven the particle distribution, the more serious the erosion. The greater the relative velocity of the particles, the more serious the erosion on the surface. The erosion of the double-suction centrifugal pump under small flow conditions mainly includes the inflow impact erosion, the friction erosion in the blade channel, the vortex erosion, and the impact erosion at the leading edge. The total erosion rate of blade wall is positively correlated with the flow rate, and increases with the increase in flow rate. With the increase in flow rate, the inflow is gradually uniform, and the inflow impact erosion and vortex erosion disappear. The erosion at the impeller is mainly the friction erosion in blade channel and the impact erosion at the leading edge.

(2) The vortex and the secondary flow in the impeller have great influence on particle trajectory. Under the influence of the flow passage vortex in the impeller, the particle trajectory moves from the suction to the pressure surface at the outlet of the impeller. The vortex increases the particle concentration at the exit section of the pressure surface of the blade and intensifies the impact erosion in this area. The particles carried by the local wall-attached vortex rub against the wall of the local vortex region, which aggravates the local friction and erosion. The greater the radius of vortex core, the greater the rotational speed of vortex and the higher the erosion rate. The secondary backflow existing in the blade channel makes it easier for the particles to separate from the carrier current. As the local velocity increases, the particles will also accelerate. This results in a higher impact velocity and an increase in the number of particles hitting the wall of the blade and the rear cover, which in turn leads to serious erosion on the wall.

(3) Since the size of the particle does not change, its force state also does not change. The particle concentration has little influence on the shape and position of erosion, but

has obvious influence on the erosion rate and erosion area, which is consistent with the phenomenon of particle movement trajectory. The greater the concentration of particles, the greater the contact area between the particles and the wall, which results in greater erosion on the wall.

**Author Contributions:** Software, X.S.; validation, D.Q.; formal analysis, Y.S.; investigation, L.X.; resources, W.W.; data curation, Y.L.; writing—original draft preparation, X.S.; writing—review and editing, Z.W. All authors have read and agreed to the published version of the manuscript.

**Funding:** This work was supported by water conservancy science and technology projects in Ningxia Hui Autonomous Region (DSQZX-KY-01, DSQZX-KY-02), the joint open fund of Tsinghua University Ningxia Yinchuan Water Network Digital Water Control Joint Research Institute (sklhse-2021-Iow10), and the National Natural Science Foundation of China (51876099).

**Institutional Review Board Statement:** Not applicable.

**Informed Consent Statement:** Not applicable.

**Data Availability Statement:** Not applicable.

**Conflicts of Interest:** The authors declare no conflict of interest.

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
