# Peer review of "Numerical Simulation Prediction of Erosion Characteristics in a Double-Suction Centrifugal Pump"

_processes, doi:10.3390/pr9091483_

Round 1

Reviewer 1 Report

  • The paper is well written and formatted according to MDPI guidelines. However, grammatical errors were observed in few instances. It is recommended to proofread the paper again with the help of a native English speaker.
  • It is recommended to break down the sentences in lines 243-246. 
  • In figure 3, the outlet section looks too short to obtain a stable flow at the pump outlet. Consider extending the volute casing for a uniform flow at the outlet. 
  • It is recommended to calculate the GCI value for the grid independence test. 
  • Why is the erosion rate higher in Qd for hub2 and not for hub1?
  • Figure 5 is a graph of the Head vs Flow rate but the title is mentioned as efficiency curve. Please correct. 
  • The labels in figure 10 are not easily readable. Consider giving a bright background for the texts. 
  • The references lack the latest researches carried out in this field. Please include recent publications. 

Author Response

Dear Editors and Reviewers:

Thank you for your letter and for the reviewers’ comments concerning our manuscript entitled “Numerical simulation prediction of erosion characteristics in a double-suction centrifugal pump” (ID: processes-1300919). Those comments are all valuable and very helpful for revising and improving our paper, as well as the important guiding significance to our researches. We have studied comments carefully and have made correction which we hope meet with approval. Revised portion are marked in the paper. The main corrections in the paper and the responds to the reviewer’s comments are as flowing:

1.Response to comments1:

This paper has been proofread again with the help of a native English speaker.

2.Response to comments2:

The sentences in lines 243-246 has been broken down.

3.Response to comments3:

Figure 3 has been modified.

4.Response to comments 4:

Due to the steady results adopted in the calculation results, the blades on both sides are asymmetrically distributed, and the flow at the pump inlet is biased, so erosion at hub2 may be high.

5. Response to comments 5 and 6:

Figure 5 and figure 10 have been modified.

6. Response to comments 7

The references have been updated.

Reviewer 2 Report

p.82 – Change «Tabakoff» instead of «tabakoff».

p.167 – Iteration «Erosion model» twice.

p.177 – The choice of such flow rates is not clear, on the basis of which the values of 0.25, 1.0 and 1.27 of Qd were selected?

p.177 – A link to table 1 is required. There is no link in the text.

p.178 – Where did the information come from (this is the average concentration or concentration at a certain time of the year), why were these concentration values taken (are there any field data on particle concentrations) in Table 1?

p.179 – How does the graph in Figure 5 compare with the data in Table 1 (Qd and 1.27Qd are presented on the graph)? Or is it data from your other article?

p.182 –  Figure 7 - there is no scale signature, perhaps there should be "Erosion rate density" and the dimension in all figures should be displayed in the same way.

p.182 –  It is better to rephrase the names of Figures 6 and 7 according to their description in the text on p. 186-188.

p.186 – Does the declared error of 0.3% correspond to the error values for the experimental and theoretical points with Qd = 3.5 m3 / s in Figure 5?

p.187 – For the first time the term "abrasion" appears, it is better to remove and insert the familiar "erosion".

p.189 – Where is the information about the particle size taken from and why is it taken as the basic size for the calculation exactly such a small value as 0.025 μm? Is there any information on the real average particle sizes?

p.204 – In almost all figures, where there is a division of figures into a), b) and c), it is necessary to refer in the text to the figure to which it refers, for example, to p.204: “Under the condition of small flow rate (see Fig.8 (a)), the flow pattern ... "

p.212 – Figure 8b below shows eddy flows at the nominal mode, what are they caused by, is there an error in the display of the figure?

p.222 – Why is Qd a NON rated condition?

p.226 – Why, in Fig. 9a, the particle velocity, in the opinion, is greater than the flow velocity?

p.232 – Missing "and erosion MASS loss of blades ...".

p.251 – Could you clarify how the Erosion rate density was calculated, it is not very clear from the text? Perhaps the calculation formula is worth giving.

p.271 – It is not clear why only 2 blades are selected? are the averaged values for all zones of mass carryover from each of the considered blades?

p.271 – The selected scale of the Flow rate axis is not clear - the scale must be adjusted to the coordinate axis (if the scale is scaled by Qd values, then the graph will behave differently - a sharp increase after Qd) or remove the notches on the axis and bring the graph as a histogram.

p.271 – How is the minimum of the obtained function explained at the nominal operating mode of the pump? Also, the selected scale of the Flow rate axis is not clear (see above).

p.290 – Extra space, ".., causing ..".

p.296 – Why "corrosion" should be "erosion"?

p.296 – A space is missing at the beginning of a new sentence.

p.298 – No point.

p.313 – Why does the erosion rate 10-8 in the graph differ from the erosion rate 10-6 for 0.25Qd in Figure 12?

p.304, 305, 314 – "Circumferential velocity", and in Figure 15 (p.313) "tangential velocity".

p.314 – Odd space.

p.320-321 – A link to table 1 is required. There is no link in the text.

p.346, 350 – At what Qd are these figures constructed?

p.359 – In the text, where possible, it is better not to mention the same word several times in one sentence, but to replace it or remove it, for example, on p.359 "... on the erosion morphology and erosion position." correct to "... on the erosion morphology and position."

p.374 – A link to the previous article is required.

p.375 – Is the presented dependence built for only one blade? Why only for one, if you can present the averaged values for all 16 blades? Or are they represented? For what Qd is this graph? Is there any confirmation of the obtained predicted dependences of the mass carryover from the full-scale operation of the pump, because it turns out that in one year with the maximum particle concentration (the particle size is also small 0.025 μm), the mass carryover from one blade can be up to 9 kg?

p.376 – Must be supplemented with "with DIFFERENT particle concentration"

p.377 – In Conclusion, it is desirable to provide the numerical data obtained in the article.

p.466 – There is no reference [22] to this article in the text.

p.470 – There is no reference [24] to this article in the text.

Author Response

Dear Editors and Reviewers:

Thank you for your letter and for the reviewers’ comments concerning our manuscript entitled “Numerical simulation prediction of erosion characteristics in a double-suction centrifugal pump” (ID: processes-1300919). Those comments are all valuable and very helpful for revising and improving our paper, as well as the important guiding significance to our researches. We have studied comments carefully and have made correction which we hope meet with approval. Revised portion are marked in the paper. The main corrections in the paper and the responds to the reviewer’s comments are as flowing:

1. Response to comments1 and 2:

According to the comments, corrections have been made in the paper.

2. Response to comments3:

According to the maximum and minimum operating conditions that the pump can reach in the project, two typical non rated working conditions of 0.25Qd and 1.27Qd and rated working conditions Qd are selected.

3. Response to comments4:

A link to table 1 has been added in the paper.

4. Response to comments5:

This is the average concentration provided by the engineering design unit.

5. Response to comments6:

Table 1 shows the design scheme, and figure 5 shows the calculation results using clear water medium. Because the experiment is a clean water experiment. This data acquisition process is introduced in my another article.

6 Response to comments7 and 8 and 9 and 10 and 11:

Relevant contents have been revised according to comments.  All figures for the erosion rate have been displayed in the same way. "abrasion"  change to“erosion”

7 Response to comments 12:

Sediment information is provided by the Hydrological Bureau of Ningxia water resources department. Not yet disclosed. In the measurement of the particle size of the Yellow River in the literature, 0.025mm is the common average particle size of the Yellow River

8 Response to comments 13:

Relevant contents have been revised according to the comments“ In almost all figures, where there is a division of figures into a), b) and c), it is necessary to refer in the text to the figure to which it refers, for example, to p.204: “Under the condition of small flow rate (see Fig.8 (a)), the flow pattern ... "

9 Response to comments 14:

The flow velocity in the pump under the rated flow condition Qd is relatively small compared with that under the large flow rate condition 1.27Qd and due to the existence of the outlet tongue, so a small-scale vortex appears near the tongue, as shown in figure 8(b).

10 Response to comments 15-19:

Relevant contents have been revised according to the comments, see in the paper.

11 Response to comments 15-19:

The calculation formula for erosion rate has been added in the paper.

12  Response to comments 20:

"It is not clear why only 2 blades are selected? are the averaged values for all zones of mass carryover from each of the considered blades?"

In order to analyze the wear difference on both sides of the blade, the blade with the most serious erosion rate on both sides is selected for analysis.

13 Response to comments 21-26:

Relevant contents have been revised according to the comments, see in the paper.

14  Response to comments 27:

The erosion diagram with 10-6 is the area erosion, and Figure 12 is the time wear diagram, multiplied by the area, so it is 10-8.

15  Response to comments 28-34:

Relevant contents have been revised according to the comments, see in the paper.

16  Response to comments 35:

Not all blades are worn, but the most severely worn and typical blades are selected for analysis.

16  Response to comments 36-44:

Relevant contents have been revised according to the comments, see in the paper.